# OpenReview forum: "Forcing Generative Models to Degenerate Ones: The Power of Data Poisoning Attacks"
_NeurIPS.cc/2023/Workshop/BUGS — NeurIPS 2023 BUGS Poster_

### Official Review · Reviewer_yHiP · 2023-10-20
**Concerns with evaluation metrics**

**Rating:** 4
**Confidence:** 3

**Review:**

## Originality and significance
- Early work in poisoning in the text summarization and text completion settings

## Clarity
- In Figure 4, it’s surprising that on both bills and xsum that Target match is so low (“avg percentage of target phrases appearing on clean samples is <8%”). It feels a bit suspicious that even with 10% poisoned training data Clean Target Match remains under 1% for the xsum dataset. This does not make much sense and is not explained in the paper.
- Authors state that pieces triggers are “has the best trade-offs between success and stealthiness in terms of attacks on both datasets” but the difference between pieces and fixed in Fig. 5b and 5e is <1%. Not sure if that is significant enough. Additionally, Figure 4b and e show different results where fixed is the stealthiest.
- Unclear how Fig. 4e has negative values when target match is an average of indicator (0 or 1).

## Quality
- Did a good job of considering multiple trigger possibilities: fixed, floating, and pieces
- In Section 3, the authors reason about the detection of the trigger placement. For example, it is stated that fixed insertion “can be easily spotted by simple checks or even human eyes”. For all of the triggers, it would benefit the paper if the authors could provide evaluations of simple checks from NLP. For example, since the threat model assumes a pre-trained model, it can be used to check perplexity of modified text.  This evaluation would support the idea that these triggers are stealthy and hard to detect. Currently, it’s hard to evaluate whether this is the case.
- It is unclear whether the metrics are suitable to evaluate whether poisoning was successful. I’d argue even one trigger phrase is enough of a success if the phrase is more than k tokens. Using an average of trigger phrase occurrence makes it hard to reason about scales (y-axis) in each plot. Every dataset will have a different set of trigger phrases.

Minor:
- pg. 2: “attack do not work” -> “attack did not work”

## Summary
Overall, my rating reflects the fact that I do not believe the evaluation metrics properly capture “stealthiness” or “success” of the attack. This makes it hard to reason about the y-axis in the plots that use clean/poisoned target match. Additionally, conclusions (such as which trigger style is best) do not seem to be consistent across the datasets, which is not explained. Still, I believe this is an interesting research direction and could make for a good paper once concerns are resolved.

---

### Official Review · Reviewer_1HsD · 2023-10-25
**Forcing Generative Models to Degenerate Ones: The Power of Data Poisoning Attacks**

**Rating:** 6
**Confidence:** 4

**Review:**

- This paper focuses on poisoning attacks on NLG tasks, which is an important topic.
- The authors proposed new metrics to profile stealthiness and attack success.
- The authors compared the security vulnerabilities of generative LLMs using both full fine-tuning and prefix-tuning, a representative PEFT method.
- However, it lacks some discussion of defenses.

---

### Decision · Program_Chairs · 2023-10-28

**Decision:**

Accept (Poster)

**Comment:**

The paper presents an interesting topic and novelty in poisoning for text summarization/completion. I believe the paper has great value to the community. Given the fact this is a workshop submission and the goal is to foster discussions of new ideas, I'm pleased to inform that the paper has been accepted to the workshop. Please carefully consider the comments put by the reviewers and revise the paper to a better form!